An exhaustive survey of regular peptide conformations using a new metric for backbone handedness (h)

Mannige Ranjan V. ranjanmannige@gmail.com
1 Molecular Foundry, Lawrence Berkeley National Laboratory , Berkeley , CA , United States
2 Multiscale Institute , Redwood City , CA , United States
Wilke Claus
Electronic publication date: 2017 May 16
Publication date: 2017
Volume: 5
Electronic Location ID: e3327
Received 2017 Jan 6; Accepted 2017 Apr 18
Copyright: ©2017 Mannige
Copyright year: 2017
Copyright holder: Mannige
License: This is an open access article distributed under the terms of the Creative Commons Attribution License, which permits unrestricted use, distribution, reproduction and adaptation in any medium and for any purpose provided that it is properly attributed. For attribution, the original author(s), title, publication source (PeerJ) and either DOI or URL of the article must be cited.
License URL: https://creativecommons.org/licenses/by/4.0/

Keywords: Ramachandran plot, Peptide backbone, Backbone chirality, Backbone handedness, Miyazawa, Peptide helix, Protein structure

Funding: Defense Threat Reduction Agency IACRO-B0845281 Office of Science, Office of Basic Energy Sciences, of the US Department of Energy DE-AC02-05CH11231 The author received no specific funding for this work. However, the author was partially supported by the Defense Threat Reduction Agency under contract no. IACRO-B0845281. This work was done at the Molecular Foundry at the Lawrence Berkeley National Laboratory (LBNL), supported by the Office of Science, Office of Basic Energy Sciences, of the US Department of Energy under Contract No. DE-AC02-05CH11231. There was no additional external funding received for this study. The funders had no role in study design, data collection and analysis, decision to publish, or preparation of the manuscript.

==============================
The Ramachandran plot is important to structural biology as it describes a peptide backbone in the context of its dominant degrees of freedom—the backbone dihedral angles φ and ψ (Ramachandran, Ramakrishnan & Sasisekharan, 1963). Since its introduction, the Ramachandran plot has been a crucial tool to characterize protein backbone features. However, the conformation or twist of a backbone as a function of φ and ψ has not been completely described for both cis and trans backbones. Additionally, little intuitive understanding is available about a peptide’s conformation simply from knowing the φ and ψ values of a peptide (e.g., is the regular peptide defined by φ = ψ =  − 100°  left-handed or right-handed?). This report provides a new metric for backbone handedness (h) based on interpreting a peptide backbone as a helix with axial displacement d and angular displacement θ, both of which are derived from a peptide backbone’s internal coordinates, especially dihedral angles φ, ψ and ω. In particular, h equals sin(θ)d∕|d|, with range [−1, 1] and negative (or positive) values indicating left(or right)-handedness. The metric h is used to characterize the handedness of every region of the Ramachandran plot for both cis (ω = 0°) and trans (ω = 180°) backbones, which provides the first exhaustive survey of twist handedness in Ramachandran (φ, ψ) space. These maps fill in the ‘dead space’ within the Ramachandran plot, which are regions that are not commonly accessed by structured proteins, but which may be accessible to intrinsically disordered proteins, short peptide fragments, and protein mimics such as peptoids. Finally, building on the work of (Zacharias & Knapp, 2013), this report presents a new plot based on d and θ that serves as a universal and intuitive alternative to the Ramachandran plot. The universality arises from the fact that the co-inhabitants of such a plot include every possible peptide backbone including cis and trans backbones. The intuitiveness arises from the fact that d and θ provide, at a glance, numerous aspects of the backbone including compactness, handedness, and planarity.

Introduction

The backbone of a protein (Fig. 1A) can twist and turn into numerous conformations (folds), in part due to the amino acid sequence that the protein displays. Understanding how a backbone twists is of great importance to the field of biochemistry, since understanding the structure of a protein goes a long way towards understanding how a protein functions (Alberts et al., 2002; Berg, Tymoczko & Stryer, 2010). While the conformation of a peptide backbone is dependent on a number of parameters (bond lengths, bond angles, and dihedral angles), Ramachandran, Ramakrishnan & Sasisekharan (1963) recognized that the twist of a peptide backbone can be described to a great degree by the dihedral angles ϕ and ψ (Fig. 1A).

Figure 1 The backbone of a single residue (A) can be described by its dihedral angles ϕ and ψ (and in smaller part, ω, which is predominantly trans or ∼180°).

The Ramachandran plot is important because a number of regular conformations important to biology—secondary structures—are located at specific regions of the plot (B). For the most part, regular peptide backbones twist in either a left-handed or right-handed fashion; examples are shown in (C). As evidenced in (B), the −ve diagonal within the Ramachandran plot (dashed line described by ϕ =  − ψ) divides right-handed peptides from left-handed peptides, which leads to the naïve picture of handedness (D). Zacharias & Knapp (2013) showed that this picture is over simplistic, however an in-depth characterization of the backbone in all regions was not performed, and will be done here for both cis (ω = 0) and trans backbones (ω = π). (A) is modified from Mannige, Kundu & Whitelam (2016). Due to low incidence within the studied database (see Methods), the two left-handed helices in (B) are arbitrarily marked and have no statistical significance. All molecular representations in this text are shown in ‘licorice’ form, with the colors red, blue and white representing oxygen, nitrogen and carbon atoms.

Today, two-dimensional (ϕ, ψ) plots are called Ramachandran plots (or ‘maps’), and are introduced in undergraduate biology textbooks as a guide for understanding a peptide backbone’s general conformational state or ‘twistedness’ at a glance (Bragg, Kendrew & Perutz, 1950; Pauling & Corey, 1951b; Pauling, Corey & Branson, 1951; Linderstrøm-Lang, 1952; Laskowski et al., 1993; Chothia et al., 1997; Hooft, Sander & Vriend, 1997;Cooper & Hausman, 2013; Alberts et al., 2002; Laskowski, 2003; Ho, Thomas & Brasseur, 2003; Eisenberg, 2003; Berg, Tymoczko & Stryer, 2010; Mannige, Kundu & Whitelam, 2016). The Ramachandran plot is especially useful because (stable) proteins are hierarchical in structure (Linderstrøm-Lang, 1952): the final (tertiary) conformation of a structured protein is composed of discrete secondary structures—regular structures—that interact with each other and which are strung together by loops that are less regular (Alberts et al., 2002; Berg, Tymoczko & Stryer, 2010). Each regular peptide structure describes a backbone whose per-residue (ϕ, ψ) values are generally the same, and therefore their ‘locations’ on the Ramachandran plot act as structural landmarks (Fig. 1B).

So far, our understanding of the Ramachandran plot has been limited mostly to structured proteins that display stable conformations (Berman et al., 2000; Alberts et al., 2002). These types of proteins occupy only a limited region of the plot (dotted regions in Fig. 1B). The regular backbone conformations in these regions are well understood. For example, known regular structures that are to the right of the negatively sloping diagonal (dashed line in Fig. 1B; henceforth denoted as the ‘ −ve diagonal’) are left-handed in backbone twist, while those that are to the left of the diagonal are right-handed (left- and right-handed regions are respectively shaded blue and red). For example, the position of the idealized left- and right-handed α-helices (Fig. 1C)—respectively denoted as αL and α in Fig. 1B—are on opposite sides of the −ve diagonal. The ‘naïve view’ of handedness, obtained from looking only at structured proteins, would be the expectation that the −ve diagonal neatly separates the Ramachandran plot into regions of left- and right-handedness (Fig. 1B).

However, structured proteins represent only a fraction of functional proteins. Indeed, up to 15% of mammalian proteins are completely disordered—they natively display multiple, often extended, conformations—and up to 50% of the mammalian proteins display large (>30 residue) stretches of disorder (Iakoucheva et al., 2002; Ward et al., 2004; Orosz & Ovádi, 2011; Mannige, 2014). Interestingly, when compared to structured backbones, structurally degenerate or disordered backbones occupy many more regions within the Ramachandran plot (Beck et al., 2008).

Additionally, a number of peptide mimics—especially peptoids (Sun & Zuckermann, 2013)—have been found to display novel secondary structures that occupy regions that are strictly disallowed by proteins due to steric clashes. For example, a ‘higher-order’ peptoid secondary structure—the Σ-strand (Mannige et al., 2015; Robertson et al., 2016) —is believed to sample regions of the Ramachandran plot (‘I’ in Fig. 1B) that are not permitted within natural proteins (this is because peptoid backbones lack hydrogen-bond donors). Another peptoid secondary structure—the ‘ ω-strand’ (Gorske, Mumford & Conry, 2016)—samples similarly historically uncharted regions of the Ramachandran plot (‘II’ in Fig. 1B ). Importantly, backbone twist handedness plays a crucial part in explaining these new motifs: as one goes along the backbones of these secondary structures, alternating residues display backbone twists that are equal in magnitude but opposite in handedness (for this reason, the Σ-strand is relatively linear, albeit meandering; Mannige et al., 2015; Mannige, Kundu & Whitelam, 2016).

Despite these recent discoveries of natively disordered proteins and novel peptidomimetic structures, a complete understanding of backbone conformations that stray from the ‘structured’ regions on the Ramachandran plot is missing, which impedes our ability to identify and explore such conformations. Towards filling this gap in understanding, this report outlines a detailed study of how regular backbones twist in every region of the Ramachandran plot for both cis and trans peptides. In particular, this report develops and explores a new metric for handedness (h) based on modeling a regular backbone (described below) as a helix (Shimanouchi & Mizushima, 1955; Miyazawa, 1961; Zacharias & Knapp, 2013). The metric is used to exhaustively chart the handedness of regular backbones. In doing so, this survey provides a new graphical format to explore new types of secondary structures being discovered (Mannige et al., 2015; Gorske, Mumford & Conry, 2016). Also, this survey dispels the naïve view of handedness (Fig. 1D) by showing that the distribution of handedness as a function of ϕ and ψ is more complicated than the distribution allowed by the naïve view. Finally, the results also show that the Ramachandran plot whose ϕ and ψ values range between 0°and 360°is more intuitive and visually meaningful (compared to those that range between −180°and 180°), particularly for cis backbones. This work builds on a previous report (Zacharias & Knapp, 2013) and helps complete our understanding of the ways in which a peptide backbone twists, which is a basic component of structural biology.

Methods

While angular units in this report switch between radians and degrees, their units in any particular situation may be inferred by the presence or absence of the degree symbol (°). All methods and materials required to produce this manuscript are freely available at https://github.com/ranjanmannige/backbone_chirality.

Deriving measures for backbone handedness

Numerous metrics for molecular chirality and handedness have so far been discussed (Harris, Kamien & Lubensky, 1999). For example, metrics for chirality have been introduced that focus on vector orientations (Kwiecińska & Cieplak, 2005; Kabsch & Sander, 1983; Gruziel, Dzwolak & Szymczak, 2013), optical activity (Osipov, Pickup & Dunmur, 1995), and molecular shape (Ferrarini & Nordio, 1998). However, this report will focus on a simpler metric for handedness associated with an idealized helix within which all (regular) backbone atoms of one type sit (Fig. 2; Shimanouchi & Mizushima (1955); Miyazawa (1961); Zacharias & Knapp (2013)). Here, a ‘regular’ backbone indicates that each tunable parameter within a unit or ‘residue’—say a particular dihedral angle—remains the same for all residues. Below, regular backbones are modeled in context of helical parameters that, when combined, form an intuitive metric for backbone handedness.

Figure 2 Internal coordinate (i) and helical coordinate (ii) representations of right-handed (A) and left-handed (B) regular backbones.

Internal coordinates are a function of bond lengths (e.g., v23), angles (σ2), and dihedral angles ( τ12), while helical coordinates are a function of displacement along the helical axis (d12), angular displacement in the plane perpendicular to the helical axis ( θ12) and shortest distance of an atom of type i to the helical axis (ρi). Representations are derived from Figs. 1 and 2 in Shimanouchi & Mizushima (1955).

Describing a regular backbone as a helix

Interest in how a backbone may be represented as a helix emerged shortly after the first secondary structures were introduced (Pauling, Corey & Branson, 1951; Pauling & Corey, 1951b; Pauling & Corey, 1951a). In particular, Shimanouchi & Mizushima (1955) had derived a set of equations that fit a platonic helix to the atoms within a regular backbone. While the formalisms described by Shimanouchi & Mizushima (1955) (and later on by Miyazawa (1961), discussed below) apply to repeating linear polymers of arbitrary complexity, this report focuses specifically on how peptides may be modeled. Figure 2 describes an arbitrary peptide backbone that may be represented either using internal coordinates (i) or helical coordinates (ii).

Internal coordinates are associated with stereochemical terms: bond lengths (vij) between adjacent atoms i and j, bond angles (σi) between the two bonds adjacent to atom i, and dihedral or torsion angles (τij), which involve atoms associated with the bond i − j and the two adjacent atoms. Helical coordinates (Fig. 2 (ii)) are described using measures of axial displacement between two successive atoms of the same type (d; this is related to the pitch of a platonic helix), angular displacement between two successive atoms of the same type (θ), and the radius of the helix (ρi) that hosts all backbone atoms of type i. Therefore, the single cylinder shown in Fig. 2 is too simplistic as there should be one distinct cylinder or radius per atom type.

Given that there are three backbone atoms associated with a residue (Fig. 1A), d = dn,α + dα,c + dc,n and θ = θn,α + θα,c + θc,n. Here, di,j and θi,j respectively refer to the axial and angular displacement between adjacent atoms i and j. Subscripts ‘n’, ‘ α’, and ‘c’ respectively refer to the backbone nitrogen, α-carbon and carbonyl carbon atoms (Fig. 1A). The notation used by Shimanouchi & Mizushima (1955) was in terms of matrices, which were then simplified by Miyazawa (1961) into trigonometric terms. In particular, Miyazawa (1961) noted that the total residue-residue axial displacement (d) and angular displacement (θ) may be retrieved using the following two equations.

cosθ2=cos+ϕ+ψ+ω2sinσn2sinσα2sinσc2

−cos+ϕ−ψ+ω2sinσn2cosσα2cosσc2

−cos+ϕ+ψ−ω2cosσn2sinσα2cosσc2

(1) −cos−ϕ+ψ+ω2cosσn2cosσα2sinσc2

dsinθ2=+vn,α+vα,c+vc,n sin+ϕ+ψ+ω2sinσn2sinσα2sinσc2

−+vn,α−vα,c+vc,n sin+ϕ−ψ+ω2sinσn2cosσα2cosσc2

−+vn,α+vα,c−vc,n sin+ϕ+ψ−ω2cosσn2sinσα2cosσc2

(2) −−vn,α+vα,c+vc,n sin−ϕ+ψ+ω2cosσn2cosσα2sinσc2

The ranges for d and θ, respectively, are [ − λ,  + λ] and [0, 2π) (the positive limit λ is defined by allowed values for the various internal coordinates). As above, subscripts ‘n’, ‘α’, and ‘c’ respectively refer to the backbone nitrogen, α-carbon and carbonyl carbon atom types. The dihedral angles ϕ, ψ, and ω represent the traditional symbols for backbone dihedral angles, which may be otherwise denoted as τn,α, τα,c, and τc,n(+1), respectively.

Finally, for any type of atom (say α-carbons), the radius or distance from the helical axis ρα is defined by 2ρα21−cosθ+d2=vα,c2+v c,n2+v n,α2−2v c,nvα,c cosσc+vn,α cosσn

(3) +2vα,cvn,αcosσccosσn−sinσcsinσncosτc,n

Miyazawa (1961) noted that the right-hand side of Eq. (3) is also the squared distance between adjacent atoms of the same type (denoted here as dα2 for α-carbons), which allows for a more simplified form (4) ρα=dα2−d22−2cosθ.

The distance between adjacent α-carbons (dα) is ∼3.8 Å  for trans peptides and ∼3 Å  for cis peptides. Other radii (ρc, ρn) can be obtained by cycling through (α, c, n) subscripts within Eqs. (3) and (4).1 Note that all ρi’s are functions of θ and d (along with other internal coordinates), and so one may use two of the three terms in (d, θ, ρi)— to describe the helical state of a peptide. Since there is only one d and θ per backbone (compared to three ρi’s, one per atom type), this report utilizes d and θ as the two descriptors (other discussions on this choice have also been made by Zacharias & Knapp (2013)).

Eqs. (1) and (2) may be substantially simplified (Miyazawa, 1961), given that backbone bond lengths and angles are much less ‘tunable’ when compared to dihedral angles (Ramachandran, Ramakrishnan & Sasisekharan, 1963; Improta, Vitagliano & Esposito, 2015a; Esposito et al., 2013; Improta, Vitagliano & Esposito, 2015b). In particular, most backbone bond lengths and angles display one equilibrium value (Improta, Vitagliano & Esposito, 2015a; Esposito et al., 2013; Improta, Vitagliano & Esposito, 2015b), while the backbone dihedral angles ϕ and ψ occupy a range of possible values and minima, e.g., regions in the Ramachandran plot that describe α-helices and β-sheets (Fig. 1B). With this in mind, Miyazawa (1961) set ω = π (trans) and substituted average (equilibrium) values for bond angles and lengths into Eqs. (1) and (2) to arrive at a simpler equation for trans backbones. Zacharias & Knapp (2013) published an updated version of this set of equations, which follows.2 (5) cosθ2=−0.8235sinϕ+ψ2+0.0222sinϕ−ψ2,

(6) dsinθ2=2.9986cosϕ+ψ2−0.6575cosϕ−ψ2.

This equation is especially relevant to peptides as they occur predominantly in trans conformations (ω = π). However, given the prevalence of cis backbones in peptide mimics such as peptoids (Mirijanian et al., 2014; Gorske, Mumford & Conry, 2016), for completeness, the corresponding relationships for a cis (ω = 0) backbone follows. (7) cosθ2=0.4052cosϕ+ψ2−0.4932cosϕ−ψ2,

(8) dsinθ2=2.3093sinϕ+ψ2+0.0028sinϕ−ψ2.

Note that Eq. (5) through Eq. (8) are simplifications of Eqs. (1) and (2), and are therefore prone to some limitations that are not present in Eq. (1) and Eq. (2). For example, bond lengths (Improta, Vitagliano & Esposito, 2015a) and bond angles (Esposito et al., 2013; Improta, Vitagliano & Esposito, 2015b) display some dependence on local backbone conformation. These subtle variations have great implications when dealing with a large number of residues, especially when considering bond angles. For example, when attempting to recreate a protein conformation from an original conformation’s ϕ and ψ values (ignoring deviations in ω, bond angles, and lengths), the original and recreated conformations tend to deviate dramatically due to an accumulation of errors (by up to 22 Å  in root mean squared deviation; Tien et al. (2013)). However, when studying changes in conformationally regular and local stretches of peptides, such deviations are not likely to change relevant features such as handedness and extent of twistedness. If circumstances indicate that the backbone values for bond angles and ω may be strained from their equilibrium values (e.g., due to bulky sidechains), only Eqs. (1) and (2) can be expected to faithfully (and perfectly) represent backbone features such as handedness of twist. However, the approximations of Eq. (5) through Eq. (8) are sufficient for the purposes of this report, given that this report primarily discusses features within platonic regular backbones.

On the one-to-one correspondence between (ϕ, ψ, ω) and (d, θ)

Given a particular value of ω, every (ϕ, ψ) pair points to exactly one (d, θ). However, when using Eqs. (1) and (2), one value of ω can not be replaced with a periodically equivalent version of ω (the same can be said for ϕ and ψ). For example, using ω = x + 2π instead of ω = x will maintain the magnitude of d and θ, but the signs will not remain conserved. This is because every summand in Eqs. (1) and (2) contains either a sine or cosine of [ ± ϕ ± ψ ± ω]∕2. The issue arises because of the ‘2’: even though the angle x is considered to be equivalent to the angle x + 2π, and even though cos(x + 2π) equals cos(x) (due to angle periodicity), cos([x ± 2π]∕2) = cos(x∕2 ± π) =  − cos(x∕2) (note the negative sign). Similarly, sin([x ± 2π]∕2) =  − sin(x∕2). Therefore, even though the angles ω and ω + 2π may be considered to be equivalent angles, expressions such as cos([x − ω + 2π]∕2) and cos([x − ω]∕2) are only equal in magnitude and not in sign. I.e., a one-to-one correspondence between (ϕ, ψ) and (d, θ) is only possible if one insists on specific values for ωs. For this reason, this report proposes to wrap the value of an amide backbone ω′ between [Δ, Δ + 360°) using (9) ω=ω′−Δ%360+Δ,

where % represents the modulus function, and Δ describes the start of the range [Δ, Δ + 2π). Choosing Δ =  − 90°would ensure that the distribution of both cis (ω = 0 ± 5°) and trans (ω = 180 ± 5°) will remain contiguous. Using this system, cis and trans backbones are respectively represented by ω = 0 (and not 2π) and ω = π (not −π) for trans backbones. The rest of this report assumes these values of ω for cis and trans backbones.

These points lead to the conclusion that a strict one to one-to-one correspondence between (ϕ, ψ, ω) and (d, θ) does not exist, since multiple sets of the former may be backmapped from the latter (by reconfiguring Eqs. (1) and (2)). Yet, a one-to-one correspondence may be ensured by discarding as solutions all but the one set of (ϕ, ψ, ω), whose ϕ and ψ lie within a preset range –e.g., [0, 2π) or [ − π, π) –and whose ω does not change after being wrapped by Eq. (9).

Introducing an equation for backbone handedness

The helical parameters d and θ host a wealth of information, some of which is discussed in the Results section. For the purpose of developing an equation for backbone handedness, it is only important to recognize, as was done before (Zacharias & Knapp, 2013), that θ and d together are instrumental in describing backbone handedness.

Figure 3 The handedness of a helix is a function of angular displacement θ perpendicular to the helical axis (green curved arrows) and linear displacement d along the helical axis (blue, vertical arrows).

Note that left-handed (L) and right-handed (R) backbone twists are respectively associated with the L and D chiralities within the Fisher Projection system and S and R chiralities within the Cahn–Ingold–Prelog system (Cross & Klyne, 2013); however, as discussed in the Methods section, this report makes a distinction between helix handedness and molecular chirality.

The relationship between handedness and (d, θ) is shown in Fig. 3. While θ indicates the extent to which a regular backbone curves along a helical path, the handedness of a backbone is dependent on both θ and d. This is because the sign of d provides a frame of reference for interpreting θ. In particular, if d is negative, then 0 < θ < π indicates left-handedness (Fig. 3A), while π < θ < 2π indicates a right-handed helix (Fig. 3B). However, if d is positive, then the manner in which the helix is ‘built’ reverses, and 0 < θ < π indicates right-handedness (Fig. 3C), while π < θ < 2π indicates left-handedness (Fig. 3D).

Given these relationships, this paper proposes a new metric for backbone handedness that depends on the sign of d and the value of θ: (10) h=d|d|sinθ.

The range of h is [ − 1, 1], with negative (or positive) values indicating that the overall twist of the backbone is left(or right)-handed. Also, |h| is proportional to the extent to which the backbone is twisted. Note that d∕|d| is related to the traditional sign function sgn(d), but deviates at d = 0, where the former term is undefined while the latter term is 0. Additionally, h will equal 0 if d = 0 or if θ = xπ (where x is an integer); for more on the meaning of d and θ in context of handedness and peptide geometry, please refer to the ‘Results and Discussions’ and Fig. 4 in particular.

Alternative measures of handedness

Two estimates for chirality, χ1 and χ2, used to validate the new measure of handedness h (Eq. (10)), were previously used by Kwiecińska & Cieplak (2005) and Kabsch & Sander (1983), respectively. The equations are: (11) χ1=1N∑i=2N−2vi−1×vi⋅vi+1vi−1vivi+1,

(12) χ2=1N∑i=2N−2arctan2vivi−1⋅vi×vi+1,vi−1×vi⋅vi×vi+1.

Here, N is the peptide length, i is the peptide residue number and the position of each α-carbon is Ni, with vector vk ≡ Nk+1 − Nk. The scalar component of the vector vi is denoted as vi. Eq. (11) has range [ − 1, 1]. Eq. (12), also used by Gruziel, Dzwolak & Szymczak (2013), is the dihedral angle associated with the four contiguous α-carbons (one preceding and two succeeding the residue i), and ranges between [ − π, π] radians. For both metrics, values deviating more from 0 are more chiral (or ‘twisted’ or ‘handed’), and left-handed twists are negative while right-handed twists are positive. Only α-carbon atom positions are used for the calculation.

Finally, a more backbone-agnostic metric of chirality has been introduced by Solymosi et al. (2002), which is replicated here purely for completeness: (13) χ3=4!3N4 ∑i,j,k,l∈Nvij×vkl⋅vilvij⋅vjkvjk⋅vklvijvjkvkl2vil.

χ3, of arbitrary range, is known as the chirality index G0S in Solymosi et al. (2002) and Neal et al. (2003). (i, j, k, l) are exhaustive permutations of {1, 2, …, N}. This metric qualitatively matches the values of Eqs. (11) and (12) and, while not shown, the relationship between (ϕ, ψ) and χ3 is available in the online GitHub repository.

Backbone structure generation

The metric h (Eq. (10)) is purely analytical and does not need structures to be computationally generated, since Eq. (5) through Eq. (8) that provide d and θ require only pairs of ϕ and ψ angles. However, if values for bond angles, lengths and dihedral angles are expected to deviate greatly from equilibrium values, θ and d can only be obtained from the more detailed Eq. (1) and Eq. (2), whose parameters would likely be obtained from a structure. On the other hand, as χ1 (Eq. (11)) and χ2 (Eq. (12)) work explicitly with atom positions, these metrics explicitly need the generation of structures. In order to use these metrics, peptides (poly-glycines) of arbitrary length were generated using the Python-based PeptideBuilder library (Tien et al., 2013). Analysis was performed using BioPython (Cock et al., 2009) and Numerical Python (Van Der Walt, Colbert & Varoquaux, 2011). Ramachandran plots that describe chirality (e.g., Fig. 5A) were generated using a grid spacing (in degrees) of ϕ, ψ ∈ { − 180,  − 178, …, 178, 180}.

Obtaining secondary structure statistics

Statistics about secondary structures—particularly α-helices, 310-helices and β-sheets—were identified using the DSSP algorithm (Kabsch & Sander, 1983), although the STRIDE algorithm (Frishman & Argos, 1995) provides qualitatively identical distributions. The DSSP algorithm was applied to a database of 13,760 three-dimensional protein conformations (one domain per conformation) with lower than 40% sequence identity, obtained from the Structural Classification of Proteins or SCOPe website (Release 2.06; Fox, Brenner & Chandonia (2014)). This database is currently available as: http://scop.berkeley.edu/downloads/pdbstyle/pdbstyle-sel-gs-bib-40-2.06.tgz.

Backbone chirality ≠ backbone handedness

Finally, it is important to recognize the distinction between backbone (twist) handedness and backbone (molecular) chirality. Naïvely, chirality is a simple concept: a molecular conformation is achiral if its mirror image can be superimposed onto itself, otherwise that conformation is chiral (Gold et al., 1997) (alternatively, and less commonly, achiral molecules possess inversion centers). Despite this intuitive definition, chirality has remained a confusing concept ever since its introduction (Bentley, 2010; Wallentin et al., 2009), which is possibly due to the fact that ‘context’ is very important when discussing chirality (Mislow, 2002). For example, when looking at a peptide at the residue or ‘local’ level, every amino acid (excepting glycine) is chiral due to the presence of a chiral α-carbon (its mirror image can not be superimposed onto itself). Yet, at the macromolecular level, even an all-glycine (and therefore locally achiral) peptide will display conformations that are not superimposable onto each other, and so such conformations would be chiral. Alternatively, when considering handedness, if a backbone is completely flat (say, a ring, where d = 0), handedness (h) will be undefined, and so one can not speak of handedness of the twist. Yet, the backbone may still remain chiral; e.g., cisplatin and transplatin are planar molecules that are nonetheless chiral opposites (Testa, 2013). It is for this reason that this report chooses to be careful to not claim that Eq. (10) is a metric for peptide/backbone chirality, but of peptide backbone twist handedness. However, estimates for backbone chirality (e.g., Eqs. (11) and (12)) may be used as surrogates for twist chirality to validate h (Eq. (10)), as both are related but not the same.

Results and Discussion

Relevance of θ and d

When discussing peptide backbones, two possible definitions of backbone ‘flatness’ (or linearity) are possible: flatness at a residue level and flatness at the atomic level. In the former, all atoms of the same type are coplanar (examples of atom types are the backbone nitrogens, carbonyl carbons, α-carbons, or even sidechain β-carbons). In the latter definition of flatness, all atoms within the backbone are coplanar. For the discussions below, since the residue-by-residue behavior of the peptide is of primary relevance, the former definition is chosen as the relevant scope for flatness.

As described in Fig. 3, the helical parameters d and θ respectively refer to an axial displacement along the helical axis and an angular displacement in a plane perpendicular to the helical axis. For example, d = 0 indicates a helix flattened along its helical axis (Fig. 4, Scenario 1). This means that all regular peptides with d = 0 will be ring-like at some peptide length (shown in a following figure for a range of peptides). As expected from Eq. (10), at d = 0, one can not tell how the helix was built, since coplanar peptides can not be described as either left- or right-twisting. Therefore, even though d = 0 indicates highly twisted peptides, these twists do not possess handedness. This shows up in the h metric because, at d = 0, |d|−1 is undefined.

Figure 4 Further discussion on the meaning of d and θ.

As shown in Fig. 3, axial separation d and angular separation θ between adjacent atoms of the same type combine to define handedness. The blue (dark) and red (light) shaded quadrants within the graph show the distribution of handedness as a function of d and θ. The relevant boundaries— θ = xπ (where x is a non-negative integer) and d = 0—separate the map into four quadrants of left- and right-twisting backbones (‘L’ and ‘R’, respectively). Geometric interpretations of various boundaries, discussed in the text, are shown to the top and left of the graph as three scenarios. The toroid enclosed by two solid lines (and shaded white) represents all possible conformations for trans peptides (ω = 180 ± 5°). Similarly, the region allowed for cis peptides (ω = 0 ± 5°) are bound by the two dashed contours.

Additionally Fig. 4 describes two important values for θ: eπ (Scenario 2) and oπ (Scenario 3), where e and o are even and odd integers. In particular, for any d, θ = eπ indicates zero angular displacement along the axis, which puts all atoms of the same type on the same line parallel to the helical axis (Fig. 4, Scenario 2). Similarly, θ = oπ indicates that every alternate atom (of the same type) along the backbone will be linear, and every adjacent atom will be diametrically opposite to each other (Fig. 4, Scenario 3); i.e., θ = oπ indicates that all atoms of the same type will lie on a plane that is parallel to the helical axis. In short, θ = 0 codes for backbones that are linear (optimally extended for a fixed d) and θ = π describe peptides that zig-zag along a plane perpendicular to the helical axis. Finally, as is evident in Fig. 4, θ = eπ conformations are not available to peptide backbones. Therefore, θ = oπ (e.g., π or 180°) will be the most extended type of backbone (for a fixed d). These relationships show how, a priori, the curve of a backbone with particular (d, θ) may be interpreted.

Finally, θ may serve as an important single-number metric for describing backbone configurations. Mannige, Kundu & Whitelam (2016) developed one such number—a Ramachandran number (R)—that is a structurally meaningful combination of ϕ and ψ. This number depends on the fact that structural features of the backbone (e.g., radius of gyration) vary least when one slices through the trans Ramachandran plot along negative-sloping lines that conserve ϕ + ψ (Ho, Thomas & Brasseur, 2003; Zacharias & Knapp, 2013; Mannige, Kundu & Whitelam, 2016). Interestingly, θ follows that trend too, which—in combination with the fact that regions of the Ramachandran plot are sparse (Mannige, Kundu & Whitelam, 2016)—means that θ and its derivatives (e.g., h) are universal Ramachandran numbers. The universality arises from the fact that cis Ramachandran plots do not conserve structure along lines that conserve ϕ + ψ (and so R only works for trans backbones), yet any two backbones with nearly identical θ’s will also be conserved in structure (see, e.g., Fig. 5A, top). This feature of θ will be true irrespective of the nature of the amide dihedral angle ω (Eq. (1)).

Figure 5 The handedness of an ordered trans peptide within the Ramachandran plot.

(A) displays the relationship between backbone parameters (ϕ, ψ) and the associated helix parameters of curvature sin(θ) (top; Eq. (1)) and axial displacement d (bottom; Eq. (2)). As shown in Fig. 3, the handedness of a helix is a function of these two variables (h; Eq. (10)). (B) is a map of backbone handedness (h) as a function of ϕ and ψ. The boundaries, θ = π (‘––’ ) and d = 0 (‘–⋅–’), correspond to backbones that are equally flat, but which are respectively optimally extended and curved (see discussion in text). (B) shows that the naïve expectation of handedness in a Ramachandran plot (Fig. 1D) is inaccurate. Interestingly, our naïve expectations would be upheld if one were only to have sampled regions of the Ramachandran plot dominated by known proteins (A; regions enclosed by ‘⋯⋯’ indicate 90% occupancy). An example of the behavior of one ‘slice’ of (B) is shown in (C). Each snapshot represents a peptide backbone that is either in a distinct region of handedness or at a boundary.

Handedness of trans backbones

Figure 5A describes the behavior of sin(θ) and d as a function of ϕ and ψ (assuming an all-trans backbone; ω = π or 180°). Figure 5B describes the behavior of backbone handedness (h; Eq. (10)) as a function of ϕ and ψ. This map is a complete description of the handedness of an all-trans (regular) peptide backbone. Fig. 5C describes some structures at various regions within the plot. As discussed above, d = 0 (‘✩’) indicates that each residue is at the same ‘altitude’, i.e., the helix is perfectly flat and maximally curved (at that particular θ). Note that any path on the Ramachandran plot that transitions from negative to positive d will encounter an infinitesimal region in its path where d = 0 and so h is undefined there. This, along with the recognition that d = 0 indicates highly curved backbones, means that such transitions would be concomitant with a sharp change in handedness. When θ = π, then the backbone is also flat (see ‘★’ in Fig. 5C); however, atoms of the same type lie in a single plane that is perpendicular to the helical axis (Fig. 4). In short, within the Ramachandran plot, d = 0 (‘–⋅–’) and h = π (‘– –’) code for flat backbones that are respectively either optimally curved (at a given θ) or optimally extended (at a given d). A future report will discuss how these simple rules may be combined to make conjectures about novel secondary and tertiary structures.

Figure 6A shows that the equation for h match other metrics for handedness, as interpreted by other metrics for chirality (Kwiecińska & Cieplak, 2005; Kabsch & Sander, 1983; Gruziel, Dzwolak & Szymczak, 2013). In particular, Fig. 6A displays the Ramachandran plot colored by h ((i); Eq. (10)) next to estimates calculated using χ1 ((ii); Eq. (11)) and χ2 ((iii); Eq. (12)). Each panel describes identical regions of left- and right-handedness, which is shown as a cartoon in (iv). However, given that χ1 and χ2 are estimates for chirality and not backbone handedness, their exact values differ from the primary metric for handedness (h) provided here.

Figure 6 (A) and (B) describe the handedness of backbone twists whose amide dihedral angles are trans (ω = π) and cis (ω = 0), respectively.

Column (i) describes handedness (h; Eq. (10)), which does not require structures to be computationally generated. Columns (ii) and (iii) respectively show vector-based estimates of backbone chirality— χ1 (Eq. (11)) and χ2 (Eq. (12))—which are calculated from computationally generated peptides (see ‘Methods’). Regions of left- and right-handedness are identical for all measures (i–iii). A cartoon representation of distinct regions of handedness is shown in (iv). Finally, (C) displays a range of regular cis peptide backbones with d ≈ 0. As explained in Fig. 4, d = 0 indicates a flat backbone that lies perpendicular to the helical axis, which results in ring-like peptides. Interestingly, a point in the Ramachandran plot exists exclusively for cis peptides, where d = 0 and θ = π: ϕ =  − ψ =  ± 36°(‘✩’  in (B)-(iv) and (C)).

Handedness of cis backbones

In the same vein as Fig. 6A, Fig. 6B displays h, χ1 and χ2 as a function of ϕ and ψ for all-cis regular backbones. This appears to be the first complete description of chirality of an all-cis backbone (ω = 0). Interestingly, the boundaries for d = 0 and θ = π switch in cis backbones, with the −ve diagonal and curved boundaries being caused by d and θ, respectively. Additionally, Fig. 6A reiterates the idea that cis peptides are quite different when compared to trans peptides: the regions and boundaries of left- and right-handedness within the Ramachandran plot differ for cis versus trans.

Finally, points on the cis map (ϕ =  ± 36°, ψ = ∓36°) exist where d = 0 and θ = π. An example of this, along with other d = 0 configurations, is shown in Fig. 6C for a six-residue peptide. At first glance, this appears to be contradiction, because d = 0 indicates the most curved backbone at a fixed θ, and θ = π indicates the most linear backbone at a fixed d; however, it is purely due to the nature of the cis backbone that this indeed is possible. Of course, this structure would only be possible for cyclic peptides with length two, given that any peptoid of length greater than two would result in overlapping atoms. However, such a structure (one with d = 0 and θ = π) is not possible in trans peptides, even in theory, because the boundaries associated with d = 0 and θ = π do not intersect (Fig. 6A); this is also evident in Fig. 4, where trans peptides are shown to not occupy regions of (d, θ) = (0, π), while cis peptides do.

The exhaustive survey of regular cis (ω = 0) and trans (ω = π) peptides (Fig. 6) proves that the naïve picture of chirality—that the −ve diagonal separates the right-twisting backbones from the left-twisting backbones (Fig. 1D)—is wrong. However, deviations from ω = 0 or π are evident in the Protein Databank; see, e.g., discussions by Improta, Vitagliano & Esposito (2011). This raises the question: how does varying ω through non-traditional values change the handedness landscape? Figure 7 describes Ramachandran plots that show handedness in terms of varying ω, which shows that this complicated separation of handedness in cis and trans backbones also holds for other values of ω. Therefore, the naïve expectation of handedness (Fig. 1D) is too simple, irrespective of amide dihedral angle.

Figure 7 The landscape of backbone chirality as a function of amide dihedral angle ω.

As ω is changed, the features of the landscape smoothly transform from the landscapes of ω =  ± π to ω = 0. For all values of ω, it is evident that the naïve view of chirality (Fig. 1D) is wrong: at least four distinct regions of chirality (separated by boundaries d = 0 and θ = π) are evident in each scenario. Although only five snapshots (values of ω) are shown, all integer values of ω were tested, which corroborates the fact that the naïve view of backbone handedness (Fig. 1D) is universally incorrect.

[−π, π) or [0, 2π): which frame of reference to use?

In structural biology, ϕ and ψ within the Ramachandran plot has been historically set to range between the values [ − π, π) radians (see, e.g., textbooks by Berg, Tymoczko & Stryer (2010) and Alberts et al. (2002)). However, Ramachandran, Ramakrishnan & Sasisekharan (1963) had originally used the range of [0, 2π). Today, the range [ − π, π) is used predominantly by structural biologists (Laskowski et al., 1993; Laskowski, 2003; Zacharias & Knapp, 2013), while some have turned to [0, 2π) as the norm (Némethy, Leach & Scheraga, 1966; Voelz, Dill & Chorny, 2011).

Given the periodicity of the Ramachandran plot, the two frames of reference are scientifically identical; however the value of the Ramachandran plot lies in its utility as a map: it is a map of important features of proteins relative to the various regions, quadrants, and diagonals in the map (see, e.g., discussions by Beck et al. (2008)). The Ramachandran plot’s value lies in being able to convey large amounts of information in easy to read pictograms. For that reason, switching the map from one range to another means that the two types of scientists—each used to a distinct range—will not be able to converse as seamlessly.

Therefore, the following question must arise: which range—[ − π, π) or [0, 2π) —is able to convey more information with the least amount of effort? Figure 8 shows the handedness of a trans backbone (A, B) and cis backbone (C, D) in the two frames of reference. From (A) and (B) it is evident that general trends in the map for trans backbones remain the same in both frames of reference: the negative diagonal (θ = π) locally separates right-handed regions from left-handed regions, while the curved line (d = 0)—which also separates handedness—also appears to be in generally the same regions (albeit inverted in curvature). The cis backbones, however, look dramatically different in the two frames of reference: the range [ − π, π] separates handedness in a more complicated manner (C), while, for the most part, the −ve diagonal appears to meaningfully separate handedness when the plot ranges from 0 to 2π (D). For this reason, purely when looking at handedness, and especially in the case of cis backbones, the Ramachandran plot that ranges between 0 and 2π appears to be more meaningful.

Figure 8 The two frames of reference (or ranges) for the Ramachandran plot for trans and cis backbones.

Both ranges [ − π, π) and [0, 2π) yield similar trends for trans backbones (A, B); however, for cis backbones, the latter frame of reference (D) appears to more neatly apportion the handedness of the backbone rather than the traditional frame of reference (C). As in Figs. 5 and 6, ‘– –’ and ‘–⋅–’ respectively correspond to boundaries defined by θ = π and d = 0. Also, regions bound by dotted contours indicate dominant regions within which proteins reside (p = 0.9).

A universal alternative to the Ramachandran plot

While the Ramachandran plot is useful enough to earn a place in undergraduate-level biology textbooks (Berg, Tymoczko & Stryer, 2010; Alberts et al., 2002), as discussed throughout this report, it is not easy to estimate features of a peptide backbone just from its (ϕ, ψ) angles (Fig. 9A). This prompted Zacharias & Knapp (2013) to introduce a new representation for backbone degrees of freedom in the form of a polar graph. In this polar representation, the θ is the angular coordinate (azimuth) and d is the radial coordinate. An example of one such representation is shown in Fig. 9B, with the direction of increasing θ reversed (compared to the cited report) to maintain relative positions of secondary structures within the Ramachandran plot (Fig. 9A). Zacharias & Knapp (2013) stated an additional reason for the introduction of the polar representation (Fig. 9B): θ, which is an angle and therefore periodic, can remain periodic as the angular coordinate in the graph.

Figure 9 Alternative representations of the Ramachandran plot.

While the Ramachandran plot is useful to map characteristics of secondary structures (A), it is not intuitive. For example, the relationship between the Ramachandran parameters (ϕ, ψ) and the handedness of a backbone is not obvious (see, e.g., the non-obvious distribution of left- and right-handed peptides as a function of ϕ and ψ). For this reason, Zacharias & Knapp (2013) introduced a graphical format involving the helical parameters d and θ in polar coordinate space (B), where the regions of left- and right-handedness are obvious (their format differs from (B) in that their θ increases in counter-clockwise fashion). (C), which is an extension of Fig. 4, introduces another graphical representation of backbone degrees of freedom based on (θ, d), but in Cartesian space. While both (B) and (C) are equally useful in understanding regions available to a protein, the text discusses some benefits of (C) as a universal map for exploring new conformations and secondary structures. Excepting the left-handed helices (αL-, 310L-helices; see ‘Methods’), each secondary structure has two contours signifying p = 0.5 and 0.8.

However, the format proposed by Zacharias & Knapp (2013) (Fig. 9B) is incomplete for a few reasons: (1) d < 0 peptides (the bottom-left and top-right regions of Fig. 5A, bottom) will never be observed in this map since only structures with d ≥ 0 are allowed; (2) all peptides with d = 0 (marked by ‘–⋅–’ in every preceding Ramachandran plot) will be compressed into one point at the center, even though Fig. 6C shows a range of legitimate d = 0 conformations; (3) while the graph is θ-periodic, the values for θ in peptides are constrained within one [0, 2π] period (peptides range between θ = π∕4 and 2π − π∕4; vertical dotted lines in Fig. 4); i.e., periodicity in θ is not required for the faithful representation of peptides. Fortunately, even though this system is not universal (again, since d < 0 structures are not accommodated), most conformations in globular proteins display positive d, and so the representation presented by Zacharias & Knapp (2013) is a reasonable one for most proteins with known structure.

Interestingly, Fig. 9A—which arranges the parameters θ and d along Cartesian axes—serves as both a universal and intuitive map for peptide backbone geometry. This is because: (1) as shown in Fig. 4, such maps reveal a wealth of information about the peptide backbone, (2) both positive and negative values of d are allowed (compared to Fig. 9B), due to the shift in the coordinate system from polar to Cartesian, and (3) this format accommodates every type of peptide conformation: any peptide (or its mimic) has a place in this map irrespective of whether the amide backbone is cis or trans or any other value; additionally, if the backbone is distorted, such distortions can also be accounted for since d and θ account for such distortions (Eqs. (1) and (2)). This is impossible to do using a single Ramachandran plot without making sweeping assumptions about backbone parameters that are not ϕ and ψ. The (θ, d) plot opens up the possibility for a new, intuitive, and universal kind of graphical representation as a supplement to the Ramachandran plot.

A departure from perfect regularity

So far, this report has focused on regular or simple backbone conformations, i.e., those that are formed from the same ϕ and ψ angles repeated along the backbone. This is particularly because a simple and visually intuitive correspondence exists (Figs. 3 and 4) between a regular backbone (described by myriad internal coordinates) and a helix that is described simply by (d, θ). However, there is a possibility that d and θ are useful even in isolation, when the unreasonable constraint of perfect backbone regularity is lifted. An example of such a departure from regularity follows.

Some secondary structures are characterized by the regular combination of two or more sets of [ϕ, ψ] (Pauling & Corey, 1951b; Pauling & Corey, 1951a; Armen et al., 2004; Daggett, 2006; Hayward & Milner-White, 2008; Mannige et al., 2015; Mannige, Kundu & Whitelam, 2016). For example, the Σ-strand is constructed by alternating between two backbone states (ϕ, ψ, ω) = (−A, B, 180°) and (−B, A, 180°), where A ≈ 120 and B ≈ 90 (Fig. 4 in Mannige et al. (2015)). It was found that the two states are similar in the extent to which the backbone twists, but opposite in handedness, which allows for these secondary structures to remain linear, albeit in a meandering way (Mannige et al., 2015). Equation (10) also describes these two states as opposite in handedness and similar in twist extent: the h for the two states are −0.34 and 0.51, respectively (the difference in magnitude is within the range of the standard deviation in h [0.391] for the β-sheet). Similarly, the α-sheet proposed by Pauling & Corey (1951a) is constructed by alternating between α(D) and αL backbone states, yet this motif is linear because each state describes equal but opposite handedness h =  ± 0.41. These points raise the possibility that, even in the absence of perfect backbone regularity, the values d, θ, and h may be considered to be residue-specific properties that may be combined to readily provide insights about higher order structures.

Conclusions

This report introduces a metric for backbone handedness (h) that is based on modeling the backbone as a helix (Fig. 2; Miyazawa (1961)). In particular, h, which is a combination of the helical parameters θ (angular displacement) and d (axial displacement), ranges from −1 to 1, and is negative (or positive) when the backbone twist is left(or right)-handed (with larger |h| indicating greater extent of twistedness). This metric (h) was used to characterize every regular backbone’s twist within the Ramachandran plot, for both cis and trans peptides. In doing so, this report dispels a naïve view of handedness (Fig. 1D), which states that backbone handedness in the Ramachandran plot is separated by the negative-sloped (−ve) diagonal. Interestingly, the reason for the naïve view makes senses when considering only trans peptides: the −ve diagonal (‘––’ in Fig. 5A) separates right-handed and left-handed twists if one considers only the regions dominantly occupied by structured proteins (‘dotted line’ in Fig. 5A). Plotting the backbone handedness (h) in the two common frames of reference— ϕ, ψ ∈ [ − π, π) and [0, 2π)—indicates that the less commonly used frame [0, 2π) may be more appropriate for interpreting cis backbones (Fig. 8).

The behavior of a backbone in cis and trans Ramachandran plots look dramatically different (Fig. 6), and so scientists dealing with new structures that have a combination of cis and trans backbones can not use one Ramachandran plot to faithfully describe these structures. Interestingly, the parameters θ and d combine all features (internal coordinates) of a contorting backbone, including the amide dihedral angle ω, which means that (θ, d) can describe any peptide backbone, irrespective of ω. Therefore, the Cartesian plot with θ and d as the x- and y-axis, respectively, serves as a unique plot for any peptide backbone (Fig. 9), with specific values and boundaries containing deep structural meaning (Fig. 4). These discussions, the author hopes, clarify a number of concepts associated with the Ramachandran plot, while providing new insights into how to interrogate the features of new protein and protein-like structures.

The author thanks Alana Canfield Mannige, Ronald D. Hills Jr, the editor, and the reviewer for their constructive input.

Additional Information and Declarations

Competing Interests

Author Contributions

Data Availability

1 ρc and ρn are obtained by the following subscript conversions: ( α → c, c → n, n → α) and ( α → n, c → α, n → c).

2 The values used by Zacharias & Knapp (2013), taken from Engh & Huber (1991) and Engh & Huber (2006), are: vn,α = 1.459 Å, vα,c = 1.525 Å, vc,n(+1) = 1.336 Å, σα = 111.0, σc = 117.2°, and σn = 121.7°. For reference, Miyazawa (1961) originally used the following values: vn,α = 1.470 Å, vα,c = 1.530 Å, vc,n(+1) = 1.320 Å, σα = 110.0, σc = 114.0, and σn = 123.0.

The author declares that there are no competing interests.

Ranjan V. Mannige conceived and designed the experiments, performed the experiments, analyzed the data, contributed reagents/materials/analysis tools, wrote the paper, prepared figures and/or tables, reviewed drafts of the paper.

The following information was supplied regarding data availability:

The author has uploaded a publicly accessible GitHub repository containing Python scripts that recreate all relevant data depicted within the manuscript. The repository is available here:

https://github.com/ranjanmannige/backbone_chirality.

This repository may also be downloaded as a compressed zip file at the following URL:

https://github.com/ranjanmannige/backbone_chirality/archive/master.zip.

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
