# Peer review of "An exhaustive survey of regular peptide conformations using a new metric for backbone handedness (h)"

_PeerJ, doi:10.7717/peerj.3327_

## Round 0.1 · original submission · Minor Revisions

· Academic Editor

Minor Revisions

Your manuscript has been carefully reviewed by one reviewer and myself. In general, the reviewer and I have similar comments. We think that overall your work is interesting and fits with PeerJ. However, we are requesting some revisions to presentation and style of your manuscript that will markedly improve it.

In particular, neither the title nor the introduction of your manuscript make it particularly clear what it is that you're doing in this paper. Your paper will be much more impactfull if you can address this issue. In particular, by the end of the introduction, I should have a better sense of what the problem is you're addressing and what your answer is going to be. As an example, you touch on the naive map of chirality in Fig. 1d, but leave the reader hanging on why it is incorrect. Giving some more intuitive motivation here will help.

Other comments follow below.

- p. 3: "samples regions of the Ramachandran plot (‘I’ in Fig. 1d)" Do you mean Fig. 1b? Also note a similar problem three lines down.

- p. 4: "We do this because any regular backbone can be associated with a specific helix" You're using the concept "regular backbone" before you define it (in the next section).

- p. 5: "Given that backbone bond lengths and angles are generally less tunable when compared to dihedral angles, we can keep them fixed." My lab has previously found that backbone bond angles actually do fluctuate, see e.g. https://peerj.com/articles/80/ (already cited in the paper). You may want to comment on this or soften your statement.

- p. 6: "while sin(θ) < π indicates right-handedness." Do you mean θ < π? Also, please proof-read the entire paragraph. As far as I can tell, you're stating the same inequalities for d < 0 and d > 0, when they actually should be reversed.

- p. 6: Instead of d/|d|, please use the sign function sgn(d).

- p. 10: Note that you have a typo in the PeptideBuilder reference. The first author should be listed as Tien MZ, not Matthew Z. Also, you're missing the middle initials for all authors.

- The image shown between lines 179 and 180 needs to be converted into a figure with a figure number and a caption. It is also really difficult to figure out what's going on. I would suggest to show a series of images with decreasing d, so that we can see how the structure collapses onto itself as d=0 is approached.

- l. 195: Capitalize Ramachandran.

- l. 204-207: There are several typos in these lines. Please copy-edit carefully.

- "Discussion" should be renamed into "Results", or possibly "Results and Discussion".

- The Conclusions section should be written in prose form, not as an itemized list. Also, the figure needs to have a figure number and caption. And finally, please don't introduce new results in the Conclusions. The figure you're showing in Conclusions is a new result and hence should be shown in the Results section.

Reviewer 1 ·

Basic reporting

This paper is generally well written and clear. It contains a few typos (see specific comments below) that can be easily corrected. Background is well introduced and references seem to be appropriate. There seems to be an almost 40 year gap between the earliest references and most recent ones. I assume this gap arises from a renewed interest in this topic that was left behind. I would still recommend the author to investigate if developments in the 80s and early 90s are important to reference. Figures are clear and well made, please check the use of punctuation in Figure 1 when refereeing to the panel labels (a) – (d).

- The use of inline figures like the ones in 180 and 214 is not standard and might lead to confusion. I suggest to label them and reference them in the traditional way.
- Although the title is quite catchy, I think the audience could benefit from a more descriptive title that reflects the content of the paper, for instance the fact that the paper introduces a simplified metric of chirality
- I assume that PeerJ allows the use of footnotes. In my opinion the use of footnotes is anachronic. In any case, the author uses too many of them which can be distracting to the reader. I suggest reducing them to the bare minimum and move such information in the main text.
- In figure 2, the helix radio is not shown but it is described in the manuscript. I suggest to incorporate an illustration of the radius rho in the figure.

Experimental design

The motivation for this research is well stated and the methods presented to resolve the question are sound. It would be beneficial is the author could clarify if there is an strict one-to-one correspondence between the use of internal coordinates and the helical coordinate approximation. Also, it would be important to incorporate a description or a reference discussing the maximal error when applying this transformation.

Validity of the findings

Since the metric introduced in this study is used strictly for “regular” backbones a discussion is needed on how accurate is this assumption for most structures in the PDB and especially for peptoids and cis-conformations that is the main focus of this study.

My understanding is that the fact that this chirality metric is not dependent on structural computations lies from the fact that the author is making the “regular backbone” assumption, however it is not clear how general is this assumption. The author should provide arguments why this is generally true and make sure it states that this is an important underlying assumption of the method.
In line 170, the author claims that the discrepancies between the metrics based on internal coordinates and the presented metric is due to the fact that these metrics are “estimates”. This seems to be misleading since the metric presented by the author is also an estimate given that we have the perfect helix approximation and the assumption of regular backbones. It would be probably more appropriate to estate that all these metrics are estimates that only deviate minimally and reach essentially the same conclusions.

One conclusion of Fig. 6d, is that if we use the range 0-360 degrees we could have a simplified description of chirality for cis backbones. I agree with this claim and I believe it is an interesting one, however, the figure does show small but non-negligible values of handedness at the upper left and lower right corners. Why did the author decide to exclude those regions at the time of labeling the handedness? If there is a sensible argument, the author should explicitly state it otherwise the arrows should also be placed in those regions even if this weakens the author's claims.

Additional comments

I believe this work could be a useful reference for the study of chirality in secondary structure but even more applicable to the growing field of biologically inspired molecule design. Although its applicability is reduced in common biological systems, I do believe it can be beneficial to the scientific community. If the author clarifies the questions and claims of this study, the paper would benefit and might be acceptable for publication.

Specific comments:

- Abstract: Please replace “In these lines” with “Along these lines”
- Line 34, replace “… or peptide backbone is dependent on the dihedral angles ..” with “… or peptide backbone can be described by the dihedral angles ..”
- Line 64, when you say “not well characterized” do you mean “not populated” ?

---

## Round 0.2 · Minor Revisions

· Academic Editor

Minor Revisions

I agree with the reviewer that the manuscript has very much improved but could benefit from one more round of careful proof-reading and copy-editing.

Reviewer 1 ·

Basic reporting

This new revision has extensive changes that aimed to answer the questions of the reviewers. The clarity and exposition has improved. Due to the process of answering questions, the manuscript ended up having, in my opinion, too many subheadings. I suggest to remove some that are not strictly necessary to improve flow.

Since many sections were rewritten, I suggest to do a thorough revision checking for typos and vague sentences to make sure the quality of reporting is maintained.

Experimental design

I have no further comments, the author has answered my questions satisfactorily.

Validity of the findings

The author has done a good job clarifying and explaining the limitations and the scope of the results. I think the article has improved substantially in this regard.

Additional comments

The author has made a very good job at updating figures and providing more details on parts that were a bit vague or confused before. I think the manuscript has improved with respect to the previous version.

---

## Round 0.3 · accepted · Accept

· Academic Editor

Accept

Thank you for your efforts in improving the clarity of your article. I think it has improved a lot.